# Protocol for an economic analysis of the randomised controlled trial of Improving the Well-being of people with Opioid Treated CHronic pain: I-WOTCH Study

Sheeja Manchira Krishnan ,[1] Vijay Singh Gc ,[2] Harbinder Kaur Sandhu ,[3] Martin Underwood ,[3,4] Sam Eldabe ,[5] Andrea Manca ,[2] Cynthia P Iglesias Urrutia ,[1,6] I-WOTCH team

For numbered affiliations see end of article.

**Correspondence to**
Dr Cynthia P Iglesias Urrutia;
cynthia.iglesias@york.ac.uk

## ABSTRACT

**Introduction** Over the last two decades, the use of opioids for the treatment of chronic pain in England has steadily increased despite lack of evidence of both long-term effectiveness in pain relief and significant, well-documented physical and mental adverse events. Guidelines recommend tapering when harms outweigh benefits, but the addictive nature of opioids hinders simple dose-reduction strategies. Improving the Well-being of people with Opioid Treated CHronic pain (I-WOTCH) trial tests a multicomponent self-management intervention aimed to help patients with chronic non-malignant pain taper opioid doses. This paper outlines the methods to be used for the economic analysis of the I-WOTCH intervention compared with the best usual care.

**Methods and analysis** Economic evaluation alongside the I-WOTCH study, prospectively designed to identify, measure and value key healthcare resource use and outcomes arising from the treatment strategies being compared. A within-trial cost-consequences analysis and a model-based long-term cost-effectiveness analysis will be conducted from the National Health Service and Personal Social Service perspective in England. The former will quantify key parameters to populate a Markov model designed to estimate the long-term cost and quality-adjusted life years of the I-WOTCH intervention against best usual care. Regression equations will be used to estimate parameters such as transition probabilities, utilities, and costs associated with the model's states and events. Probabilistic sensitivity analysis will be used to assess the impact of parameter uncertainty onto the predicted costs and health outcomes, and the resulting value for money assessment of the I-WOTCH intervention.

**Ethics and dissemination** Full ethics approval was granted by Yorkshire & The Humber—South Yorkshire Research Ethics Committee on 13 September 2016 (16/YH/0325). Current protocol: V.1.7, date 31 July 2019. Findings will be disseminated in peer-reviewed journals, scientific conferences, newsletters and websites.

**Trial registration number** International Standard Randomised Controlled Trial Number (49 470 934); Pre-result.

### Strengths and limitations of this study

► First economic evaluation of a complex intervention Improving the Well-being of people with Opioid Treated CHronic pain (I-WOTCH) to support opioid tapering.

► The economic analyses use patient-level information to inform a de-novo decision model.

► I-WOTCH's decision model will enable estimation of the costs, health consequences, and uncertainty associated with opioid use and tapering over a lifetime horizon.

► Valuable evidence for potential implementation of the self-management support intervention aimed at opioid tapering.

► Uncertainties may remain as to the long-term effectiveness and cost-effectiveness of the I-WOTCH intervention due to the limited follow-up duration of the current trial.

## BACKGROUND

Nearly half of the UK adult population (43%) is living with chronic pain (ie, pain lasting >3 months), the prevalence of which increases with age.[1] Opioids are commonly prescribed for chronic pain and a recent study reports an increase in the number of opioid prescriptions (approximately 34%) in England between 1998 and 2016.[2] Analysis of prescription data shows that after this long increasing trend, there is a slight decrease in the annual number of opioid prescriptions for pain since 2016.[3] Despite the demonstrated short-term effectiveness of opioids, evidence of their long-term impact in terms of pain relief and improvement in functional status is scant.[4 5] This situation is compounded by concerns over the fact that long-term use of opioids can lead to adverse events (at an

estimated absolute rate of 78% in trials using a placebo as a comparator) affecting the respiratory, cardiovascular, gastrointestinal and central nervous systems.[6] These include common adverse events such as dry mouth, nausea and constipation, as well as more serious adverse events including sleep-disordered breathing, respiratory depression and opioid-related deaths (at an absolute rate of 7.5%).[6–8] Equally concerning is the fact that opioid use is associated with mental health and anxiety disorders, major depression and dysthymia.[9] Their long-term use may cause problematic patterns of substance use, leading to substance use disorders (ie, abuse and dependence).[10] Risk factors for opioid-related adverse events include older age, higher doses of this class of drugs and their long-term use.[11 12]

Clinical guidelines for prescribing opioids in chronic pain recommend tapering when the possible harms from their use outweigh any expected benefits[13] and yet, to our knowledge, no validated protocol or intervention exists to help patients reduce their opioid doses and manage their chronic pain.[14] A number of studies have evaluated interventions (eg, acupuncture, ketamine-assisted dose reduction and behavioural strategies such as motivational interviewing, psychiatric consultation, cognitive behavioural therapy and mindfulness) that support opioid tapering.[15–18] The conclusions drawn by these studies are limited by the quality of their design or insufficient follow-up period.[14] Relevant for our study, there are no existing economic evaluations of opioid-tapering strategies in the management of chronic non-malignant pain.

Improving the Well-being of people with Opioid Treated CHronic pain (I-WOTCH) is a National Institute for Health Research-funded randomised controlled trial (RCT) evaluating the effectiveness and cost-effectiveness of a patient-centred, multicomponent self-management intervention targeting withdrawal of strong opioids among those living with chronic non-malignant pain.[19] The I-WOTCH trial is designed to help people reduce their opioid consumption, manage pain interference and enhance their quality of life.

This paper describes the protocol for the economic analysis that has been designed as an integral part of the I-WOTCH trial.

## METHODS
### Study details (population, setting, location, intervention and comparator)
A detailed study protocol for the I-WOTCH trial has been reported in a separate manuscript.[19] Briefly, I-WOTCH is a multisite, patient-centred, open RCT enrolling adult patients with non-malignant chronic pain in England. The trial's target sample size is 542 participants, individually randomised (1:1) to receive the I-WOTCH intervention or best usual care. The I-WOTCH intervention is an 8–10 weeks' course—consisting of a mixture of group sessions led by two trained I-WOTCH facilitators, two one-to-one individual sessions and two telephone consultations with the I-WOTCH-trained nurse facilitator—adjunct to best usual care. Best usual care consists of general practitioner (GP) care with relaxation package and a booklet called 'My Opioid Manager'.

### Planned start and end dates
The I-WOTCH study started in September 2016 and is expected to end in March 2021. The economic analysis is expected to begin in October 2020.

### Type of economic evaluation (cost-consequences and cost-effectiveness analyses)
Two types of economic analysis will be conducted: a within-trial cost-consequences analysis (CCA) and a long-term model-based cost-effectiveness analysis (CEA), the details of these are provided in the Analysis section.

### Study perspective
The perspective for both the CCA and CEA will be that of the National Health Service and Personal Social Service for England.

### Time horizon
The CCA and CEA will adopt a 12-month and a patients' life time horizons, respectively.

### Discount rate
Estimates of mean cost and health benefits observed during the 12-month trial follow-up period will not be discounted (for the purpose of the CCA), while those predicted to accrue beyond the study follow-up (for the purpose of the CEA) will be discounted using a 3.5% annual discount rate as per National Institute for Health and Care Excellence (NICE) guidelines.[20]

### Identification, measurement and valuation of health outcomes
Table 1 describes the variables of interest to the economic analyses of I-WOTCH, the frequency of data collection and data sources.

The CCA will consider the following three health outcome measures over the 12-month trial follow-up period:
1. *Activities of Daily Living* (ADL)—measured using the Patient Reported Outcomes Measurement Information System pain intensity short form-8A questionnaire (PROMIS-PI-SF-8A).[21]
2. *Severity of opioid withdrawal symptoms*—measured using the Short Opiate Withdrawal Scale (ShOWS).[22]
3. The *5-level EQ-5D version* (EQ-5D-5L) instrument—a preference-based generic measure of health-related quality of life typically used in healthcare economic evaluation.[23 24]

The PROMIS-PI-SF-8A comprises eight questions rated on a scale of 1–5 that aim to measure the degree of interference of pain on day-to-day activities, work around the home, ability to participate in social activities, household chores, fun activities, enjoyment of social activities, enjoyment of life and family life. The total raw score is the sum of individual responses to each question thus forming a

**Table 1**  Data collection strategy for I-WOTCH's economic analyses

| Data collected | Source | Time of collection |
|---|---|---|
| *Baseline data* | | |
| Age | PtQ | Baseline |
| Gender | PtQ | Baseline |
| Ethnic group | PtQ | Baseline |
| Current work status | PtQ | Baseline |
| Age at leaving full time education | PtQ | Baseline |
| Pain duration | PtQ | Baseline |
| Opioid intake duration | PtQ | Baseline |
| Pain conditions | PtQ<br>GPRs | Baseline<br>At 12 months only |
| Pain severity stratification group | PtQ | Baseline |
| *Measures of health benefit* | | |
| Activities of daily living (PROMIS-PI-SF-8A) | PtQ | Baseline, 4, 8 and 12 months |
| Severity of opioid withdrawal symptoms (ShOWS) | PtQ<br>PtD | Baseline, 4, 8 and 12 months<br>Weekly over first 4 months |
| Generic health-related quality of life (EQ-5D-5L) | PtQ<br>PtD | Baseline, 4, 8 and 12 months<br>Weekly over first 4 months |
| *Resource use (volume, admissions, consultations, attendances and/or contacts)* | | |
| Medications | PtQ<br>GPRs | Baseline, 4, 8 and 12 months<br>At 12 months only |
| Hospital | PtQ<br>GPRs | Baseline, 4, 8 and 12 months<br>At 12 months only |
| Hospital outpatient | PtQ<br>GPRs | Baseline, 4, 8 and 12 months<br>At 12 months only |
| GP surgery | PtQ<br>GPRs | Baseline, 4, 8 and 12 months<br>At 12 months only |
| GP home | PtQ<br>GPRs | Baseline, 4, 8 and 12 months<br>At 12 months only |
| GP telephone | GPRs | At 12 months only |
| Practice nurse | PtQ<br>GPRs | Baseline, 4, 8 and 12 months<br>At 12 months only |
| Practice nurse telephone | GPRs | At 12 months only |
| District nurse (ie, at home) | PtQ<br>GPRs | Baseline, 4, 8 and 12 months<br>At 12 months only |
| NHS 111 | GPRs | At 12 months only |
| Occupational therapist | PtQ | Baseline, 4, 8 and 12 months |
| Counsellor | PtQ | Baseline, 4, 8 and 12 months |
| Psychologist | PtQ | Baseline, 4, 8 and 12 months |
| Social worker | PtQ | Baseline, 4, 8 and 12 months |
| Physiotherapist | PtQ | Baseline, 4, 8 and 12 months |
| Referrals | GPRs | At 12 months only |
| Investigations | GPRs | At 12 months only |
| Ambulance incidents | GPRs | At 12 months only |
| Accident and emergency | GPRs | At 12 months only |
| Other | PtQ<br>GPRs | Baseline, 4, 8 and 12 months<br>At 12 months only |

EQ-5D-5L, 5-level EQ-5D version; GP, general practitioner; GPRs, GP records; I-WOTCH, Well-being of people with Opioid Treated CHronic pain; NHS, National Health Service; PROMIS-PI-SF-8A, Patient-Reported Outcomes Measurement Information System pain interference-short form-8A; PtD, patient diary; PtQ, patient questionnaire; ShOWS, Short Opiate Withdrawal Scale.

maximum score of 40 and a minimum score of 8. Higher scores reflect large interference or change in participant's ability to perform daily activities. This instrument supports calculation of a common metric and will be converted to a T score.[21]

ShOWS identifies the severity of opiate withdrawal symptoms on 10 different categories namely: feeling sick, stomach cramps, muscle spasms, feeling of coldness, heart pounding, muscular tension, aches and pains, yawning, runny eyes and insomnia. The response to each question is attributed to an individual score of 0–3, with 0 indicating no symptoms, 1 mild, 2 moderate and 3 severe.[22] Higher overall scores (estimated as the sum of all individual scores with a maximum score of 30 and a minimum score of 0) indicate higher severity of opioid-withdrawal symptoms.

The EQ-5D-5L Questionnaire[23] describes health in five domains (mobility, self-care, usual activities, pain or discomfort, and anxiety or depression). Each domain has five levels of severity (1—no problems/2—slight problems/3—moderate problems/4—severe problems/5—unable to do). A response consists of a sequence of five digits, for example, 12315, 12112, which represent the level of severity on each domain reported by the respondent. Combinations of the levels of the five domains describe 3125 possible health states. Several valuation studies have been carried out in the literature to estimate value sets for a given country/region. These studies used methods consistent with economic theory to elicit the respondent's preferences towards the health states defined by the EQ-5D. A value set to calculate utility values for the EQ-5D-5L has been published recently[25] but is still subject to methodological controversy. Until the controversies are resolved, we will convert EQ-5D-5L responses onto the EQ-5D-3L scale using the mapping function developed by van Hout *et al*[26] following current NICE's recommendation.[24]

The CEA will integrate predicted survival and EQ-5D index scores (for each model state and clinical event) to derive an estimate of quality-adjusted life years (QALYs)—the health benefit of choice in our CEA—under the I-WOTCH and best usual care strategies.

### Identification, measurement and valuation of resource use

As described in table 1, patient-level resource use data are collected in the trial and will be complemented by information routinely collected in GP records. Costing will be carried out in UK pound sterling at 2019 prices. Unit cost for tests, investigations, inpatient hospital admissions and day care procedures will be estimated using National Reference Costs and Healthcare Resource Group codes.[27] Referrals and consultations will be costed using Personal and Social Services Research Unit (PSSRU) statistics.[28] If necessary, we will use unit costs from previous versions of the PSSRU report[29 30] and inflate them to the year 2019 using inflation and price indices from the Office of National Statistics (ONS).[31] Costs associated with usual care will include the cost of the relaxation CD (ie, printing and production costs) and 'My Opioid Manager'.

### Microcosting of I-WOTCH intervention

A microcosting of the resources required to provide the I-WOTCH intervention will be conducted. A detailed description of the categories of resource use to be considered, their associated unit costs and the data collection forms used in the I-WOTCH microcosting exercise is provided in the online supplemental tables A1–4. Salary, facility and travel costs will be assumed to be independent of the number of participants. Salaries for the facilitators and trainers will be estimated based on average daily salary by grade. The facility costs will be calculated based on the number of venues hired, number of days hired and daily venue hire rate. Travel costs will be considered as fixed cost per mile. Unit costs for nurses' time for face-to-face and telephone consultations will be extracted from the PSSRU 2019.[28] A per participant, locality and course cost of I-WOTCH intervention will be reported.

### Cost of medication

Unit costs of the medications (table 1) will be obtained from the British National Formulary (BNF).[32] For each strength and preparation of a given pain killer drug, we will extract the cost per pack (or bottle) from the BNF and we will calculate the relevant morphine equivalent dose (MED) using the same algorithm used in the I-WOTCH clinical analysis. This is being updated from that used to estimate MED for stratification at the time of randomisation. For each opioid-based medication reported to have been used by individuals in the I-WOTCH study, we will estimate their unit cost per MED and use it to estimate a weighted average cost per MED over all opioid medications used in the trial.

### Modelling

A Markov state-transition model was developed to facilitate the estimation of the long-term mean costs and QALYs associated with the I-WOTCH intervention and usual care. The initial conceptual structure of the model was informed by a systematic analysis of the components of the I-WOTCH intervention. This task was supported and enhanced by critically appraising published decision models that evaluated the use of opioids in chronic non-malignant pain. The face validity of the model structure was further refined by holding a series of meetings with the project team involving clinical experts from the I-WOTCH study. This process is described in detail in a separate related manuscript currently in preparation.

Figure 1 depicts a schematic representation of our final model structure, which is organised around five key states (ie, represented as ovals in figure 1). At any time period, patients can be in one of the following mutually exclusive states: (a) long-term opioid therapy (LTOT)—representing individuals candidate for the intervention, who have been using strong opioids for more than 3 months; (b) I-WOTCH tapering (IT)—individuals committed

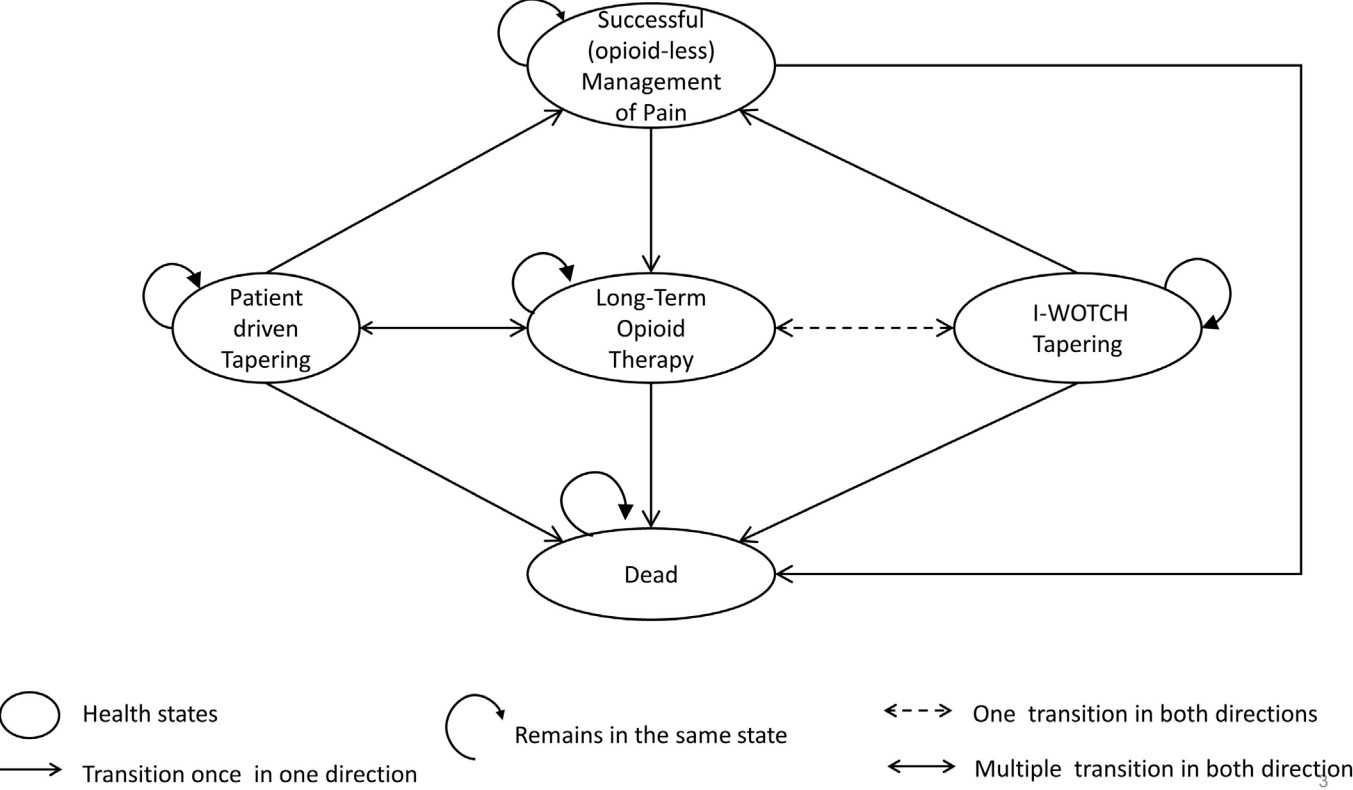

**Figure 1** Simplified model structure showing patient flow during the I-WOTCH trial. At any time, individuals will be allocated to any five health states (shown as ovals). Individuals start in the long-term opioid therapy (LTOT) health state. Depending on the trial arm, they will move to either I-WOTCH tapering (IT) or patient-driven tapering (PT). The transition from LTOT to IT occurs only at the start of the trial and patients cannot go back to IT once they withdraw from it. Patients who have tapered their opioid doses completely will be in the successful opioid-less management of pain (SOLMP) health state. Patients may withdraw from IT or PT to move to LTOT health state. Similarly, patients in SOLMP may move to LTOT if they restart their opioid doses. The arrows' spearheads indicate the direction of allowed transition from one state to the other. Dead (D) state is an absorbing state where no transitions from the D state are allowed.

to tapering as part of the I-WOTCH intervention; (c) patient-driven tapering (PT)—individuals self-committed to tapering without the intervention; (d) successful opioid-less management of pain (SOLMP)—individuals completed withdrawal from opioid use; and (e) dead (D).

Each state is associated with an average cost and utility score, derived using the methods described in the Statistical analysis section. Spearheads of arrows in figure 1 represent the directions of allowed transitions per cycle. LTOT is the starting state for all patients in the I-WOTCH trial. Individuals in LTOT can engage with IT only once but are able to initiate PT at their will. While they remain in the LTOT state, individuals can experience opioid-related transient/emergent adverse events or persistent/serious adverse events. Occasionally, adverse events may trigger a transition from LTOT to PT. Individuals in IT or PT may withdraw from opioid tapering at any point in time and go back to LTOT. Individuals in IT or PT can transit to SOLMP state only when they are no longer taking any opioids. Finally, individuals are at risk of death at any point in time (ie, transitions into the absorbing D state are allowed from the LTOT, IT, PT and SOLMP states). A paper providing a detailed description of the

conceptualisation of the model, health states definition, clinical events and associated transitions is in preparation. To maximise the use of the data collected during the trial and to model the events and transitions observed during the treatment and follow-up period, the state transition model uses a weekly cycle length during the first 4 months and monthly cycles beyond that.

## ANALYSIS
Statistical analysis for the CCA and CEA will be conducted using I-WOTCH's individual patient data (IPD) collected on resource use and outcomes, analysed on an intention-to-treat (ITT) basis. Statistical analysis and decision models will be implemented in R.[33]

### Within-trial cost-consequences analysis
The CCA will focus on estimating and reporting mean health outcomes, resource use and costs by treatment arm at each follow-up interval. No incremental analysis will be conducted. Total costs will be the sum of the costs of the healthcare resource items (described in table 1) that patients used during the study period, plus the cost

of either the I-WOTCH intervention or the usual care. Mean health outcomes will include those described in the Identification, measurements and valuation section and be reported at each follow-up period by treatment arm.

### Handling missing data

We will assess the extent of missing data in the patient-level costs and health outcomes collected during I-WOTCH's 12-month follow-up and apply appropriate methods to handle the issue following the recommendations by Faria *et al*.[34]

### Statistical analysis

Descriptive statistics (ie, mean, SD, lower and upper quartiles, minimum and maximum values) for all continuous variables reported in table 1 plus total costs will be estimated by trial arm. Histograms and/or box-plots will be used to represent these variables graphically. Binary and categorical variables will be represented in terms of percentages. The impact of patient's baseline characteristics (ie, age, gender, pain severity, opioid use, duration of opioid intake and opioid medication-related pain condition) as predictors of outcomes will be explored in a series of regression models fitted to ADL, ShOWS, EQ-5D-3L utility scores and cost data to inform the estimation of the Markov model's input parameters.

We will map EQ-5D-5L utility scores into EQ-5D-3L utility scores.[24 26] To account for idiosyncrasies of the EQ-5D-3L (ie, multimodality, truncated support and left-skewness), we will use a series of regression models (including mixture of beta models, adjusted limited dependent variable mixture models, two-part models). Each model goodness of fit and assessment of their predictive ability will be carried out using the methods recommended by Hernandez *et al*.[35 36]

Many methods have been used to analyse cost data.[37] We will use generalised linear models (with Gaussian or gamma distributed errors and identity or log links) and where necessary, two-part extensions of these models to account for any mass at zero, the right skewed nature of the dependent variable and possible heteroskedasticity. Should administrative censoring (due to patients' differential follow-up) be present, the analysis will use inverse probability weighting methods alongside our cost-regression models.[38]

All regression models will explore the impact of the patient's baseline characteristics included in table 1, and use these results to explore the role of patients' heterogeneity onto the cost-effectiveness results (more in the Subgroup analysis section).

### Sensitivity analysis

For the purposes of the CCA, we will carry out a per-protocol analysis and compare the results from the ITT. We will calculate the minimum versus actual number of group sessions needed to provide the I-WOTCH intervention to all participants in the intervention group. We will also consider a minimum and maximum number

of participants per course observed in the I-WOTCH trial. We will use a one-way or threshold analyses on the assumptions made on estimating the costs.

### Long-term model-based cost-effectiveness analysis

#### Populating model parameters

Table 2 provides a list of key model parameters and the source for initial values, that is, mean, SD or CI. To estimate all model initial values, we will use IPD from the I-WOTCH trial and GP records as well as any relevant publicly available evidence. Transitions from all model states to death will be extracted from age and sex adjusted all-cause mortality statistics from the ONS.[39] Transition probabilities between all the other model states will be derived using risk equations from regression models (eg, time-to-event and/or logit-regression models depending on the nature of the transition). Costs and utilities parameters for each of the model states and events will be estimated using the regression models described in the Statistical analysis section. Variance–covariance matrices for each of the regressions will be extracted to inform the parameter uncertainty estimates in the model and used in the probabilistic sensitivity analysis (PSA) as explained in the Probabilistic sensitivity analyses section.

### Incremental analysis

We will estimate the differential costs and QALYs predicted by the model and—where relevant—calculate the incremental cost-effectiveness ratio (ICER), defined as the ratio between the mean difference in costs and the mean difference in QALYs. As decision makers need to assess whether an intervention is 'value for money', we will compare the ICER against two 'thresholds': one ranging between £20 000 and £30 000 per QALY gained to mimic the criterion used by NICE for its policy decisions, and another recently proposed by Lomas *et al* who estimated this threshold to be between £5000 and £15 000.[40]

### Subgroup analysis

In order to reflect variation between individuals in terms of their health benefit, resource use and costs, we will run the Markov state transition model for a series of different patient profiles. This will enable us to reflect the impact that patient characteristics may have on the value for money assessment of I-WOTCH. The rationale for this approach stems from the recognition that the baseline risk (and possibly the treatment effect) may vary between individuals based on their opioid use, pain severity and pain conditions. The trial's predefined subgroups based on baseline variables are as follows: (1) pain severity score (5–8, 9–11 and 12–15); (2) opioid usage in MED (0–20 mg, 30–59 mg, 60–89 mg, 90–119 mg, 120–149 mg and >=150 mg); (3) pain conditions (fibromyalgia, musculoskeletal, arthritis, back pain, neurological, cancer and other) and (4) duration of opioid intake (less than 1 year, 1–5 years and more than 5 years).

**Table 2** Model parameters to inform long-term decision analytic model

| Transition probability | Sources | Specific details of the fields in source |
|---|---|---|
| Remaining in IT | NTP<br>NCRF | Time of withdrawal from IT<br>Time of withdrawal from IT |
| LTOT to IT | The trial | Proportion of people engaged in IT |
| IT to LTOT | NTP<br>NCRF | Time of withdrawal from IT<br>Time of withdrawal from IT |
| IT to SOLMP | NTP<br>NCRF | Time of completely stopping the use of opioids<br>Time of completely stopping the use of opioids |
| IT to dead state | PL | All-cause mortality data from ONS |
| Remaining in LTOT | The trial | Number of people who do not engage in IT and remain in LTOT |
| SOLMP to LTOT | PtQ<br>GPRs | Medication data<br>Prescription data |
| LTOT to PT | PtQ<br>GPRs | Medication data<br>Prescription data |
| LTOT to dead state | PL | All-cause mortality data from ONS |
| Remaining in PT | The trial<br>PtQ<br>GPRs | Number of people who remain in PT over time<br>Medication data<br>Prescription data |
| PT to LTOT | PtQ<br>GPRs | Medication data<br>Prescription data |
| PT to SOLMP | PtQ<br>GPRs | Medication data<br>Prescription data |
| PT to dead state | PL | All-cause mortality data from ONS |
| Remaining in SOLMP | PtQ<br>GPRs | Medication data<br>Prescription data |
| SOLMP to dead state | | All-cause mortality rates from ONS |
| **Utility scores** | **Source** | **Specific details of the source** |
| LTOT | PL<br>PtQ | Utility of opioid therapy[45]<br>Regression analysis of patient self-reported EQ-5D |
| PT | PtQ | Regression analysis of patient self-reported EQ-5D |
| IT | PtQ | Regression analysis of patient self-reported EQ-5D |
| SOLMP | PtQ | Regression analysis of patient self-reported EQ-5D |
| **Costs** | **Source** | **Specific details of the source** |
| LTOT | PL<br>PtQ<br>GPRs | Cost of opioid therapy per cycle[46]<br>Costs associated with self-reported resource use<br>Costs associated with resource use collected |
| PT | The trial<br>PtQ<br>GPRs | Cost associated with usual care<br>Costs associated with self-reported resource use<br>Costs associated with resource use collected |
| IT | The trial | Cost associated with intervention |
| SOLMP | PL | Assumption—1× contact with GP[28] |

'The trial' means the I-WOTCH trial.
GP, general practitioner; GPRs, GP records; IT, I-WOTCH tapering; I-WOTCH, Improving the Well-being of people with Opioid Treated CHronic pain; LTOT, long-term opioid therapy; NCRF, nurse clinical record form; NTP, nurse tapering plan; ONS, Office of National Statistics; PL, published literature; PT, patient-driven tapering; PtD, patient diary; PtQ, patient questionnaire; SOLMP, successful opioid-less management of pain.

### Probabilistic sensitivity analyses

Sampling uncertainty in the model will be captured by characterising each model parameter with an appropriate probability distribution. We will derive the parameters to inform these distributions from the variance–covariance matrix obtained from each regression model. Cholesky decomposition of each variance–covariance matrix will be used to make the simulation more efficient. Sampling uncertainty will be propagated through the model using Monte-Carlo simulation in the form of PSA to understand

the effect on the predicted mean QALYs and mean costs associated with each treatment arm. These parameters will then be combined and compared against a range of feasible cost-effectiveness thresholds to obtain an estimate of the probability that I-WOTCH is cost-effective. This information will be represented graphically in the form of a cost-effectiveness acceptability curve.[41] Also, we will perform scenario analyses for the various distributional assumptions made, for example, on the parametric survival distribution and compare the results for different assumptions.

## DISCUSSION

The scale of the opioid crisis in the UK is not as severe as in the USA. However, recently, it has been reported that opiates are a frequent cause of death due to drug poisoning in the UK.[3 42] To begin reviewing the benefits and risks of opioid medicine, and make recommendations for regulatory action, an expert working group has been formed by the Medicines and Healthcare Products Regulatory Agency.[43] This protocol describes the first economic evaluation of an adjunct intervention (ie, I-WOTCH) to support opioid tapering in patients with chronic non-malignant pain. Its findings will provide timely and significant results to inform policy recommendations on how best to tackle the opioids epidemic in the UK and manage the complex landscape of opioids-related health (and financial) risks.

A study that tested the effectiveness of an earlier version of I-WOTCH (ie, the COPERS trial/COping with persistent Pain, Effectiveness Research into Self-management) showed sustained benefits on depression and anxiety in people with musculoskeletal chronic pain.[44] The COPERS intervention was designed based on cognitive behavioural therapy principles. While in the COPERS trial more than 20% of participants used strong opioids, this was not aimed at opioid dose reduction. In contrast, the I-WOTCH intervention was designed to support opioid reduction and improve activities of daily living.

## Strengths

This paper describes the first economic evaluation analysis of a complex intervention to support opioid tapering, which uses data collected alongside the first UK-based RCT. A within-trial CCA will allow estimating relevant model parameters. The long-term impact on costs and health benefits from implementing the I-WOTCH Programme in England will be assessed in a lifetime decision analytic model populated using trial and previously published data. This analysis is based on a robust conceptual model that reflects clinical practice and long-term adverse events associated with both opioid use and opioid withdrawal. This model will enable estimation of I-WOTCH's cost-effectiveness over a lifetime horizon and real-life subgroup analyses.

## Limitations

Our decision problem and associated model structure are complex. I-WOTCH's effectiveness in the short and long term is associated to the occurrence/absence of several interlinked events that determine individual's transition from one health state to the other. I-WOTCH's 12-month trial follow-up period is unlikely to allow to capture all events of interest in the long term. This may limit our ability to estimate all relevant transition probabilities on trial-based IPD. A number of model parameters may need to be estimated from published sources and experts' opinion. Data availability may limit the successful evaluation of the CEA model.

## ETHICS AND DISSEMINATION

The study was approved by the Yorkshire & The Humber—South Yorkshire Research Ethics Committee (16/YH/0325). Appropriate local approvals were sought for each area in which recruitment was undertaken. The current protocol version is 1.7 date 31 July 2019. To inform all health technology assessment stakeholders, our results will be published in peer reviewed journals and presented at scientific conferences. Similarly, I-WOTCH's newsletter and lay summaries of our results on the study's website will be our main vehicles to disseminate our findings to study participants and facilitators.

## PATIENT AND PUBLIC INVOLVEMENT

The involvement of patient and public in the intervention design, development and delivery is outlined in the I-WOTCH clinical protocol paper.[19] In brief, two lay advisers with chronic pain withdrawal of opioids and experience of clinical trial research have been recruited to the study. Additionally, prior to receiving funding for the study, meetings were held with volunteers with chronic malignant pain to receive input to the design of the intervention structure, duration of intervention and content to be covered. These meetings contributed also to the design of the study including randomisation, best usual care intervention, recruitment processes, as well as outcome measures.

**Author affiliations**
[1]Department of Health Sciences, University of York, York, UK
[2]Centre for Health Economics, University of York, York, UK
[3]Warwick Clinical Trials Unit, Warwick Medical School, University of Warwick, Coventry, UK
[4]University Hospitals Coventry and Warwickshire NHS Trust, Coventry, UK
[5]Pain Department, James Cook University Hospital, Middlesbrough, UK
[6]Danish Center for Healthcare Improvements, Aalborg University, Aalborg, Denmark

**Acknowledgements** The authors would like to thank members of the I-WOTCH trial team for their comments on previous versions of I-WOTCH economic analysis plan and acknowledge the inputs of the investigators contributing to the conduct of the trial.

**Collaborators**  I-WOTCH team: (chief investigator)—Harbinder Kaur Sandhu, Warwick Clinical Trials Unit, University of Warwick. (co-chief investigators)—Sam Eldabe, The James Cook University Hospital, Cheriton House, Marton Road Middlesbrough; Charles Abraham, School of Psychological Sciences, Faculty of Medicine, Dentistry and Health Sciences, University of Melbourne, Victoria 3010, Australia; Sharisse Alleyne, Warwick Clinical Trials Unit, University of Warwick; Shyam Balasubramanian, Department of Anaesthesia and Pain Medicine University Hospitals Coventry and Warwickshire NHS Trust; Lauren Betteley, Warwick Clinical Trials Unit, Warwick Medical School; Katie Booth, Warwick Clinical Trials Unit, University of Warwick; Dawn Carnes, Barts & The London Queen Mary's School of Medicine and Dentistry; Andrea Dompieri Furlan, Toronto Rehabilitation Institute, University Health Network, Canada; Kirstie Haywood, Division of Health Sciences, University of Warwick; Maddy Hill, Warwick Clinical Trials Unit, University of Warwick; Ranjit Lall, Warwick Clinical Trials Unit University of Warwick; Andrea Manca, Centre for Health Economics, University of York; Dipesh Mistry, Warwick Clinical Trials Unit, University of Warwick; Vivien Nichols, Warwick Clinical Trials Unit, Warwick Medical School, University of Warwick; Jennifer Noyes, The James Cook University Hospital, Cheriton House, Marton Road Middlesbrough; Anisur Rahman, Centre for Rheumatology Research, University College London, London; Kate Seers, Warwick Research in Nursing, Warwick Medical School, University of Warwick; Jane Shaw, The James Cook University Hospital, Cheriton House, Marton Road Middlesbrough; Nicole Tang, Department of Psychology University of Warwick; Stephanie Taylor, Barts & The London Queen Mary's School of Medicine and Dentistry; Colin Tysall, University/User Teaching and Research Action Partnership, University of Warwick; Martin Underwood, Warwick Clinical Trials Unit, University of Warwick, Coventry; Emma Withers, Warwick Clinical Trials Unit, University of Warwick; Cynthia Urrutia, Centre for Health Economics, University of York.

**Contributors**  SMK drafted the first version of this manuscript, CPIU is responsible for overseeing the design and implementation of the I-WOTCH economic analysis and for drafting subsequent versions of the manuscript, VSG contributed to the review of the literature of economic evaluation studies. AM provided expert advice and contributed to the draft of the manuscript. HKS and SE are co-chief Investigators and oversee the running of the trial. MU has provided input into all aspects of the trial study design and support in running of the study. SMK, VSG, CI, HKS, MU, SE and AM reviewed and approved the final version of the paper.

**Funding**  This project is funded by the National Institute of Health Research (NIHR), Health Technology Assessment (HTA) (project number 14/224/04).

**Disclaimer**  The views and opinions expressed therein are those of authors and do not necessarily reflect those of HTA, NIHR or NHS.

**Competing interests**  MU was the chair of the NICE accreditation advisory committee until March 2017 for which he received a fee. He is the chief investigator or co-investigator on multiple previous and current research grants from the UK National Institute for Health Research, Arthritis Research UK, and is a co-investigator on grants funded by the Australian NHMRC. MU is an NIHR senior investigator and has received travel expenses for speaking at conferences from the professional organisations hosting the conferences. He is a director and shareholder of Clinvivo that provides electronic data collection for health services research. MU is part of an academic partnership with Serco related to return to work initiatives. He is a co-investigator on a study receiving support in kind from Stryker. MU has accepted honoraria for teaching/lecturing from the consortium for advanced research training in Africa. He is an editor of the NIHR journal series, and a member of the NIHR Journal Editors Group, for which he receives a fee. SE is an investigator on number of NIHR and industry-sponsored studies. SE received travel expenses for speaking at conferences from the professional organisations organising these conferences. SE attended advisory boards and provided consultancy services for Medtronic, Abbott, Boston Scientific, and Mainstay Medical, none in relation to opioids. SE's department has received research funding from Medtronic. HS is the director of Health Psychology Services, providing psychological services for a range of health-related conditions. AM has received consultancy fees for participating in Pharmaceutical and Medical Device Industry Advisory Boards in the area of musculoskeletal pain. AM is a member of the NICE Technology Appraisal Committee. CPIU is a member of the NICE Medical Technologies Advisory Committee.

**Patient consent for publication**  Not required.

**Provenance and peer review**  Not commissioned; externally peer reviewed.

**ORCID iDs**
Sheeja Manchira Krishnan http://orcid.org/0000-0001-8574-695X
Vijay Singh Gc http://orcid.org/0000-0003-0365-2605
Harbinder Kaur Sandhu http://orcid.org/0000-0003-1522-8078
Martin Underwood http://orcid.org/0000-0002-0309-1708
Sam Eldabe http://orcid.org/0000-0002-9250-1886
Andrea Manca http://orcid.org/0000-0001-8342-8421
Cynthia P Iglesias Urrutia http://orcid.org/0000-0002-3426-0930

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
