## [Reviewer comments · BMJ Open]

ARTICLE DETAILS

TITLE (PROVISIONAL)	Protocol for an economic analysis of the randomised controlled trial of improving the wellbeing of people with opioid treated chronic pain: I-WOTCH study
AUTHORS	Manchira Krishnan, Sheeja; Gc, Vijay Singh; Sandhu, Harbinder Kaur; Underwood, Martin; Eldabe, Sam; Manca, Andrea; Iglesias, C

VERSION 1 – REVIEW

REVIEWER	Rajiv Sharma Portland State University United States of America
REVIEW RETURNED	13-May-2020

GENERAL COMMENTS	This paper provides an excellent description of the economic analyses planned in conjunction with the I-WOTCH trial. The economic analyses will comprise a cost-consequences analysis for health outcomes/costs observed over the 12-month duration of the trial as well as a cost-effectiveness analysis with a time horizon that extends to the subjects' lifetimes. I found the paper to be clearly written and well organized. Key elements of the data and the analytical approach are described succinctly. These descriptions are valuable to other researchers who may be considering economic analyses of cost and outcomes data from clinical trials. My only suggestion for improvement is a thorough re-read of the paper to correct minor errors in writing/typing, and to ensure that the terminology used by the paper is either widely-understood or clearly defined. The first paragraph had a couple of instances where clarification may be useful. For example, I assume that "~34%" is meant to be read as "approximately 34%". I was also a little puzzled by the term "absolute rate" where the authors probably meant "combined rate" or "total rate". Again, these suggestions are very minor in the context of an excellent paper.
--

REVIEWER	Ceri Phillips Professor of Health Economics and Head of College of Human and Health Sciences, Swansea University
REVIEW RETURNED	18-May-2020

GENERAL COMMENTS	Thank you for the opportunity to review this paper. It is very well written and conforms to recognised conventions for economic
---

	evaluations of 'complex' interventions. The authors highlight the potential limitations in their proposed approach and therefore the only other query I have relates to the paper in preparation, which describes a detailed description of the conceptualization of the model, health states definition, clinical events and associated transitions. I appreciate that it is the topic for another paper, but some more detail of the model etc. would be of value to this paper as well.
--	--

VERSION 1 – AUTHOR RESPONSE

Reviewer: 1

4. This paper provides an excellent description of the economic analyses planned in conjunction with the I-WOTCH trial.

Response :
Thanks

5. I found the paper to be clearly written and well organized.

Response :
Thanks

6. Key elements of the data and the analytical approach are described succinctly. These descriptions are valuable to other researchers who may be considering economic analyses of cost and outcomes data from clinical trials.

Response :
We agree and very much hope our efforts in this sense can help others design their health economics studies.

7. My only suggestion for improvement is a thorough re-read of the paper to correct minor errors in writing/typing, and to ensure that the terminology used by the paper is either widely-understood or clearly defined.

Response :
Thank you for this suggestion. We have asked all co-authors to proof-read the manuscript to i) identify any additional typos and ii) make sure that the terminology used is either widely-understood or clearly defined.

8. The first paragraph had a couple of instances where clarification may be useful. For example, I assume that "~34%" is meant to be read as "approximately 34%". I was also a little puzzled by the term "absolute rate" where the authors probably meant "combined rate" or "total rate".

Response :
Many thanks for these comments. We have corrected the text to read "approximately 34%." Also, we went back to the relevant source reference to verify the statement on the "absolute rate" of opioid-related adverse events. We confirmed that the source reference refers to an "absolute rate of adverse events in trials using a placebo as a comparator." Thus, for clarity, we modified our statement to read

“at an estimated absolute rate of 78% in trials using a placebo as a comparator.” Please see paragraph 1 on page 4.

9. Again, these suggestions are very minor in the context of an excellent paper.

Response :

Thank you again for such a positive review.

Reviewer: 2

10. Thank you for the opportunity to review this paper. It is very well written and conforms to recognised conventions for economic evaluations of 'complex' interventions.

Response :

We appreciate the positive feedback

11. The authors highlight the potential limitations in their proposed approach and therefore the only other query I have relates to the paper in preparation, which describes a detailed description of the conceptualization of the model, health states definition, clinical events and associated transitions. I appreciate that it is the topic for another paper, but some more detail of the model etc. would be of value to this paper as well.

Response :

We appreciate it may be difficult to assess a protocol paper without the fine details of the subsequent model. Here we have done our best to provide as much detail as possible within the constraints of the word limits imposed by the journal. We are planning on submitting the model conceptualisation paper very soon and would be happy to share a copy, strictly in confidence, with the referee if that helps.